# Effect of Variable Conditions of Exposure on the Physical and Mechanical Properties of Blockboards

Octavia Zeleniuc and Camelia Coşereanu *

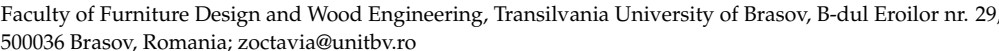

Faculty of Furniture Design and Wood Engineering, Transilvania University of Brasov, B-dul Eroilor nr. 29, 500036 Brasov, Romania; zoctavia@unitbv.ro
* Correspondence: cboieriu@unitbv.ro

**Abstract:** The growing demand for wood and the continued increase of the raw material price have resulted in companies using more efficient wood resources in wood-based products such as blockboard. These boards are used in the field of interior design, especially. The aim of this research was to evaluate the effect of variable environmental conditions on the physical and mechanical properties of blockboard (BK). Two types of commercial BK were exposed in variable environmental conditions (kitchen, bathroom and climatic chamber) for three months. The BK structures were composed of veneer (V) and high-density fibreboards (HDF) for the face sheets and glued, solid wood, resinous strips for the core. The temperature and relative humidity of the air (RH) ranged from 18 °C to 25 °C and from 25% to 90%, respectively. In the climatic chamber (CC), the dynamics of the parameter variations were different than in the other two exposure climates, being determined by the programmed cyclic changes in RH. The changes observed on physical and mechanical properties were more extreme for the blockboards tested in the CC than in the kitchen (K) and bathroom (Ba). After 3 months of exposure in the CC, the thickness and weight of the boards increased by 1.8 and 1.1 times, respectively, for veneered BK, and by 4.4 and 0.4 times, for BK with HDF faces, compared to values recorded in the kitchen. The panels exposed in the CC exhibited the highest increase in moisture content (MC): 41% for veneered BK and 82% for BK with HDF faces after three months of exposure, followed by those exposed in the bathroom and kitchen. Increases in MC determined decreases of flexural properties and soundness surfaces, more evident for HDF face structures compared to V ones.

**Keywords:** blockboard; environmental conditions; moisture content; physical-mechanical properties; dimensional stability; warpage

## 1. Introduction

Wood-based composites are comprised of a large diversity of boards, including the blockboard category. Blockboard is considered to have a plywood structure due to its design of at least three layers (usually thin veneers for the faces and thick ones for the core). The veneers are oriented perpendicular to the core grains' orientation. The thickness of the face layers ranges from 2 mm to 3.5 mm and they are glued to the core under high pressure. However, blockboard differs from plywood as the core is made of strips of solid timber (resinous or softwood species) cut from low-grade logs or short timber, contributing, thus, to the more efficient use of wood and wood-waste recovery. Total production of plywood, blockboards and LVL together represented 40% of all wood-based panel production in 2018 with a production of 163 million m³ [1], of which 145,000 m³ represents only the blockboards production of Holzindustrie Schweighofer, the world's largest blockboard plant [2].

In recent years, there has been a growing demand for wood, and the price of logs has experienced a huge increase. For this reason, some of the timber industry has been forced to efficiently use existing wood resources, including the wood with defects or small-sized

wood resulting from wood processing, fast-growing wood species [3] or recycled wood, such as post-consumer wood [4] or wood from construction sites [3,5].

In the furniture industry, there has been an increase in the demand for the manufacture of blockboards due to their structural and strength advantages. Blockboard is preferred over plywood for furniture manufacturing, especially in applications such as long shelves, top tables and side panels, because of the high resistance to warping or twisting and its durability. Blockboard is a highly appreciated, sustainable material used mainly for furniture, combining its durability with light weight. The actual research work aims to find solutions to lower the weight of blockboard by using light cores from resinous wood with various machined profiles and resistant face layers [6]. The blockboard panels are preferred by the furniture manufacturers because they can replace some solid wood elements with this material, lowering the product weight and maintaining the same properties and quality. Blockboard is also an ecological material with a very low formaldehyde emission level [7,8].

Most studies were oriented to the study of physical and mechanical performances of blockboards constructed with various cores and faces. A part of the research works was oriented to the possibility of recycling wood resulting from construction sites for the core of blockboard with and without bonding and investigating the influence of core material on the mechanical properties of the boards [5,9]. Innovative methods without adhesive, such as dowel welding of blockboard core strips with 0–20° dowel insertion angle, resulted in good mechanical properties of the boards [9]. A part of the studies investigated new designs of the structures [6,10] and various raw materials for the core [11–13]. Other studies were oriented to the joint types used for the core stripes, to their lengths and their influence on the properties of blockboards [9,14,15].

There are few studies related to blockboard properties and structures. The influence of moisture content or of the relative humidity of air on the physical and mechanical properties of wood-based composites were also in the attention of researchers, but only in the case of particleboards and fibreboards [16–18]. There are also studies with regard to weathering stability and durability of plywood [19] and studies on lightweight, stabilised blockboard panels, a newly introduced product on the market, that show a decrease in internal bond and bending strength with the increase of the relative humidity of the air from 30% to 85% [6].

The present research aims to analyse the effect of variable environmental conditions on the physical and mechanical properties of blockboards in order to assess the drawbacks of using this material in spaces where an increase in humidity could influence their stability and mechanical strengths. Two types of blockboards (with veneer and HDF faces) were exposed in three different climate conditions (in kitchen and bathroom spaces and in a programmed climatic chamber environment), and, after three months, their behaviour was evaluated. The surface warpage through the evolution of deflection was also evaluated.

## 2. Materials and Methods

### 2.1. Materials

The blockboard samples with 18 mm thickness and sizes of 500 mm × 250 mm (length × width) were supplied by Holzindustrie Schweighofer (Comănești, Romania). The blockboard structure supplied by Holzindustrie Schweighofer was composed of three layers, including spruce strips of 25 mm width for the core and the two face sheets made from various materials. The strips were, first, edge-to-edge glued, and, after that, the two face sheets were glued and applied to the core under high pressure conditions.

The face sheets of the blockboards used in the present research work were made from fromager (*Ceiba pentandra*) veneer (V) and high-density fibreboard (HDF) of 3 mm thickness (Figure 1). The adhesive used for bonding the faces to the core was urea-formaldehyde adhesive (UF) (called IF20), while the softwood strips for the core were bonded with polyvinyl acetate dispersion (PVAc D3). Blockboard boards from which the specimens were cut for the experiments had the following characteristics: board format: 2500 mm × 1250 mm; moisture content: 8–12%; thickness tolerance: +0.2 mm−0.6 mm; emission class: E1.

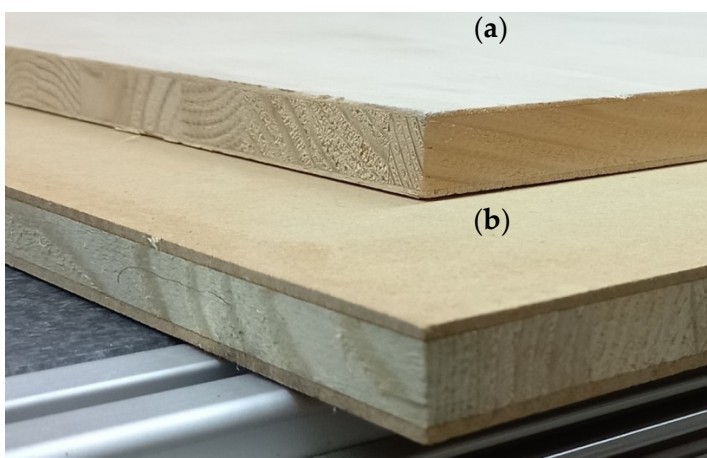

**Figure 1.** Blockboard structure: (**a**) panel with veneer (V) face sheets; (**b**) panel with HDF face sheets.

### 2.2. Design of the Experimental Conditions

To evaluate the performance of blockboard in different climate conditions, in terms of temperature and relative humidity of air (RH), three exposure situations were designed for the present experiment: indoor spaces, including both apartment kitchen and bathroom places, and programmed cyclical RH changes in a climatic chamber. The exposure conditions are presented in Table 1.

**Table 1.** The exposure conditions for the blockboard panels.

| Environment Exposure | Temperature Range, °C | Relative Humidity of Air (RC), % | Test Duration, Months |
|---|---|---|---|
| Kitchen | 18–25 | 25–64 | 3 |
| Bathroom | 18–25 | 43–88 | 3 |
| Climatic chamber | 20–25 | 50–90 * | Cyclic test of 5 days, repeated 12 times in 3 months |

\* The relative humidity was set as follows: day 1: 50%; day 2: 60%; day 3–4: 90%; day 5: 65%.

The specimens used for the experiments were given code numbers, as presented in Table 2. The boards sized at 500 mm × 250 mm × 18 mm were divided into three groups depending on the exposure situation. Six specimens were used for each group and each structure. The termohygrometer was used to register the temperature and relative humidity of the environment.

**Table 2.** The samples identification.

| Environment Exposure | Kitchen | Bathroom | Climatic Chamber |
|---|---|---|---|
| V structures | K-V | Ba-V | CC-V |
| HDF structures | K-HDF | Ba-HDF | CC-HDF |

### 2.3. Physical and Mechanical Characterisation

Sampling for determination of physical and mechanical properties was performed in accordance with [20]'s standard. The thickness of the boards was measured with a caliper with an accuracy of 0.1 mm on six distinct points (one point in the middle of the short edges and two points on the long edge situated at equal distances to the mid-point of the edge).

The thickness, weight and moisture content were measured every week, and, after each month, the test samples were cut from a part of the exposed blockboards, and physical

and mechanical properties were evaluated and compared with the properties determined before testing.

The number and sizes of the test samples used for physical and mechanical evaluation were according to the European standards: EN 322:1996 for moisture content [21], EN 323–1996 for density [22], EN 318:2002 for thickness swelling [23], EN 310:1993 for bending strength (MOR) and modulus of elasticity (MOE) [24] and EN 311:2003 for surface soundness [25]. The dimensional stability of the panels caused by RH changes was investigated by measuring the thickness variation and using the OPTOdesQ Measurement Table (Hecht, Germany). The thickness (g) was measured at 15 points in order to evaluate the deflections and warpages of the specimens after being exposed to the three different environmental conditions. It should be mentioned that the panels were measured at the same 15 distinct points set on the surface at each time of evaluation (Figure 2).

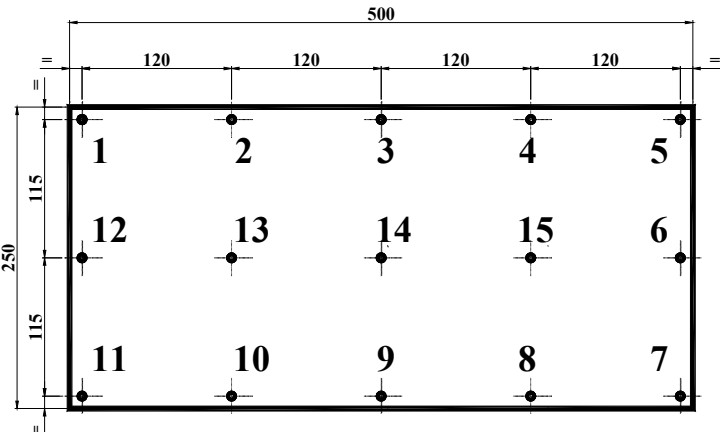

**Figure 2.** The 15 measuring points of deflection.

The mechanical tests were carried out on a universal testing machine (Zwick/Roell Z010, Germany) equipped with a ±10 kN load cell. The samples were conditioned at $20 \pm 2$ °C and $65 \pm 5\%$ relative humidity of air until they reached equilibrium.

## 3. Results and Discussion

### 3.1. Physical Properties

Hygroscopic surfaces of wood composites as blockboards which are exposed to indoor environment absorb moisture when RH increases and desorb moisture when the RH decreases. This behaviour has influence on the properties of boards. An analysis on the physical transformations of such boards must include the effect of variable environmental conditions of exposure on thickness (t), weight (w) and moisture content (MC). The environmental conditions include the variation of temperature (T) and the relative humidity of air (RH). These parameters were measured twice a day (days*2) for the entire 90-day period of testing and their variations for kitchen and bathroom conditions of exposure are presented in Figure 3.

The relative humidity (RH) varied between 43% and 88% in the bathroom and between 25% and 64% in the kitchen. The temperature (T) fluctuated from 18.2 °C to 25.6 °C in the kitchen and from 18 °C to 26.4 °C in the bathroom. Correspondingly, the equilibrium moisture content of panels in the bathroom during exposure was far higher (9.2% to 17.5%) than in the kitchen (6.1% to 11.2%), being influenced by the sudden increases in the relative humidity caused by the activities carried out. Exposure to such a variation of climate led to changes of the thickness (t) value, moisture content (MC) and weight (w) of the boards, as presented in Table 3.

The density before exposure varied between 387 kg/m$^3$ and 446 kg/m$^3$ for panels with veneer (V) faces and between 403 kg/m$^3$ and 573 kg/m$^3$ for those with HDF faces.

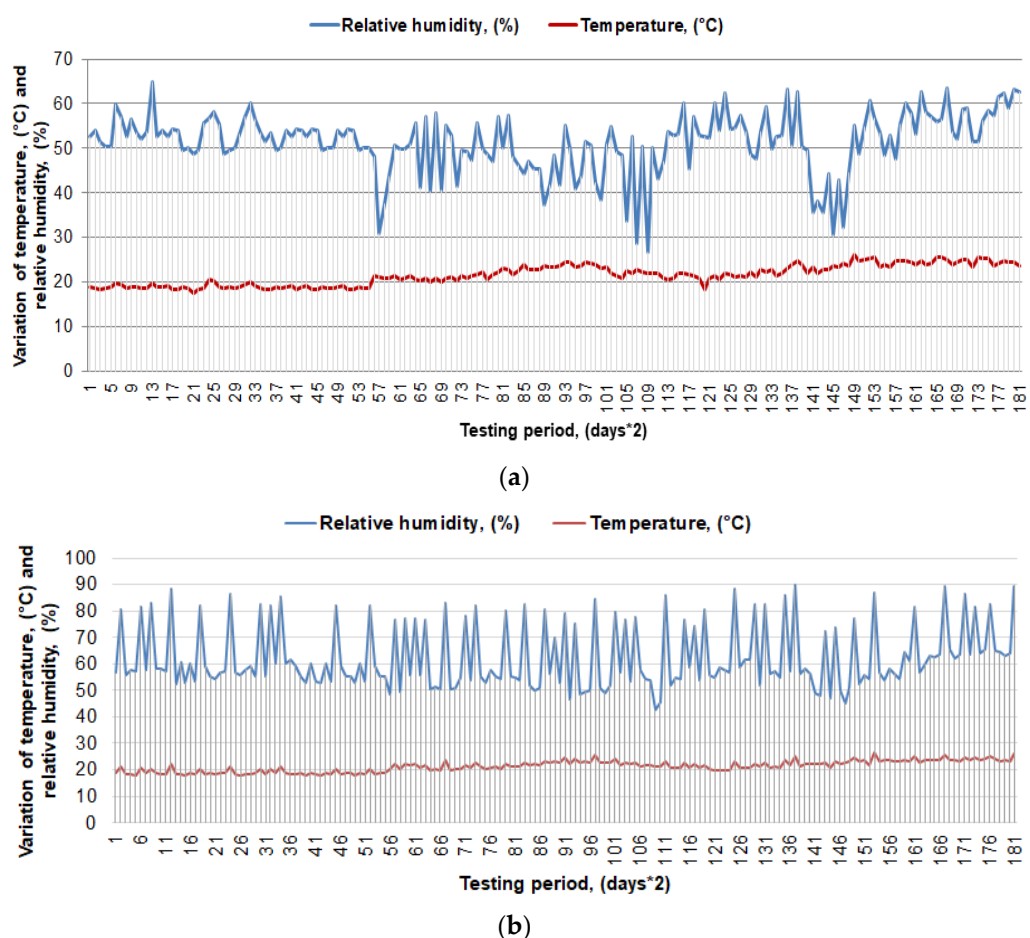

**Figure 3.** Variation of the environmental parameters: (**a**) in the kitchen; (**b**) in the bathroom.

**Table 3.** Variation of the blockboards' physical parameters during their exposure in the kitchen conditions (average values recorded for each month).

| | K-V | | | K-HDF | | |
|---|---|---|---|---|---|---|
| | t, (mm) | w, (g) | MC, (%) | t, (mm) | w, (g) | MC, (%) |
| Before testing | 18.11 (0.09) | 971 (3.5) | 8.07 (0.3) | 17.68 (0.09) | 1293 (2.5) | 6.7 (0.02) |
| 1 month | 18.13 (0.06) | 986 (1.2) | 8.50 (0.4) | 17.75 (0.11) | 1315 (3.9) | 7.1 (0.4) |
| 2 month | 18.15 (0.06) | 992 (4.4) | 9.20 (0.4) | 17.70 (0.09) | 1320 (5.3) | 7.7 (0.3) |
| 3 month | 18.25 (0.09) | 988 (3.9) | 9.6 (0.5) | 17.82 (0.11) | 1342 (6.0) | 9.9 (0.9) |

Values in the parenthesis are standard deviations.

It can be seen that there were no sudden variations from one month to another; the changes being more evident after three months, especially MC. After three months, the thickness of panels increased by approximately 0.77% (K-V) and 0.79% (K-HDF) and the weight by 1.75% (K-V) and 3.8% (K-HDF), with slight differences between blockboards with veneer and HDF face sheets. MC increased by approximately 17% for V blockboards and by approximately 48% for HDF ones, which recorded, before exposure, the lowest MC. The blockboards with HDF faces exhibited a higher increase of MC after two months of exposure. This can probably be attributed to the moisture homogenisation in the mass of the fibres.

The differences between recorded values from one month to another in the case of blockboards tested in the bathroom (Table 4) were more evident than for those exposed in the kitchen. The increases in thickness and weight were 0.64 and 0.52 times higher for V blockboards and 0.9 and 0.21 times higher for HDF blockboards compared to those exposed in the kitchen after 3 months of exposure. The high values of the relative humidity of air recorded in the bathroom (between 50% and 88%) caused more rapid moisture uptake than in the kitchen, even if, after one month, slight decreases were observed in thicknesses and weights. Nevertheless, after 3 months, the increases in MC (19% for V blockboards and 52% for HDF blockboards) were comparable with those obtained in the kitchen. These could be attributed to the dynamics of swelling and shrinkage of panels in the environmental conditions with high variability of RH. MC increased slower in HDF structures compared to veneered ones, but, in time, the fibres absorbed more humidity from the atmosphere and, thus, reached a higher MC after 3 months.

**Table 4.** Variation of the blockboards' physical parameters during their exposure in the bathroom conditions (average values recorded for each month).

|  | Ba-V | | | Ba-HDF | | |
|---|---|---|---|---|---|---|
|  | t, (mm) | w, (g) | MC, (%) | t, (mm) | w, (g) | MC, (%) |
| Before testing | 18.12 (0.05) | 938 (1.6) | 8.35 (0.2) | 17.68 (0.09) | 1305 (2.5) | 6.98 (0.02) |
| 1 month | 18.21 (0.08) | 959 (2.2) | 8.46 (0.5) | 17.81 (0.11) | 1331 (3.9) | 7.37 (0.4) |
| 2 month | 18.18 (0.07) | 955 (2.5) | 8.83 (0.3) | 17.80 (0.09) | 1328 (5.3) | 7.9 (0.3) |
| 3 month | 18.35 (0.09) | 963 (5.7) | 9.95 (0.4) | 17.95 (0.11) | 1321 (6.0) | 10.6 (0.9) |

Values in the parenthesis are standard deviations.

In the climatic chamber, the dynamics of the parameter variations were different than in the other two exposure climates (Table 5), being determined by the programmed cyclic changes in RH. After three months of exposure, the thickness and weight increases were 1.8 and 1.1 times higher for V blockboards and 4.4 and 0.4 times higher for HDF blockboards compared to values recorded in the kitchen. The highest thickness swelling (TS) values were achieved by blockboards with HDF faces in the climatic chamber (4.29%) and the lowest ones by blockboards with V face sheets in the kitchen (0.77%) after three months of exposure. TS values recorded for the tested boards in the kitchen and bathroom increased after one month by 0.11% and 0.49%, respectively, for V blockboards and by 0.39% and 0.73% for HDF blockboards. After two months, slight decreases were registered, indicating lower RH changes in the environment. Cyclic changes in RH had an effect on the TS increase in the climatic chamber after one month with 1.82% for V blockboards and 3.6% for HDF blockboards. These increases continued until the end of the exposure period. TS values were much higher for blockboards with HDF face sheets compared to those with V face sheets at the end of the exposure period. It can be attributed to the low moisture diffusion in the veneer faces with transverse grain orientation, which inhibits deeper moisture penetration: a fact also observed in plywood by other researchers [26].

**Table 5.** Variation of the blockboards' physical parameters during their exposure in the climatic chamber (average values recorded for each month).

|  | CC-V | | | CC-HDF | | |
|---|---|---|---|---|---|---|
|  | t, (mm) | w, (g) | MC, (%) | t, (mm) | w, (g) | MC, (%) |
| Before testing | 18.16 | 948 | 8.05 | 17.72 | 1153 | 6.54 |
|  | (0.06) | (3.5) | (0.5) | (0.07) | (6.5) | (0.8) |
| 15 days | 18.47 | 986 | 9.23 | 18.25 | 1211 | 8.83 |
|  | (0.09) | (6.4) | (0.4) | (0.1) | (8.4) | (0.8) |
| 1 month | 18.49 | 994 | 10.65 | 18.37 | 1226 | 10.5 |
|  | (0.08) | (8.2) | (0.3) | (0.15) | (10.9) | (1.0) |
| 2 months | 18.52 | 998 | 11.01 | 18.40 | 1232 | 11.06 |
|  | (0.07) | (8.9) | (0.9) | (0.14) | (10.9) | (1.1) |
| 3 months | 18.55 | 984 | 11.37 | 18.48 | 1211 | 11.88 |
|  | (0.09) | (9.2) | (0.8) | (0.12) | (8.9) | (0.9) |

Values in the parenthesis are standard deviations.

The accelerated weathering in climatic chamber more rapidly influenced the moisture uptake by all blockboards (Figure 4). The higher the RH, the higher the observed increase in moisture content.

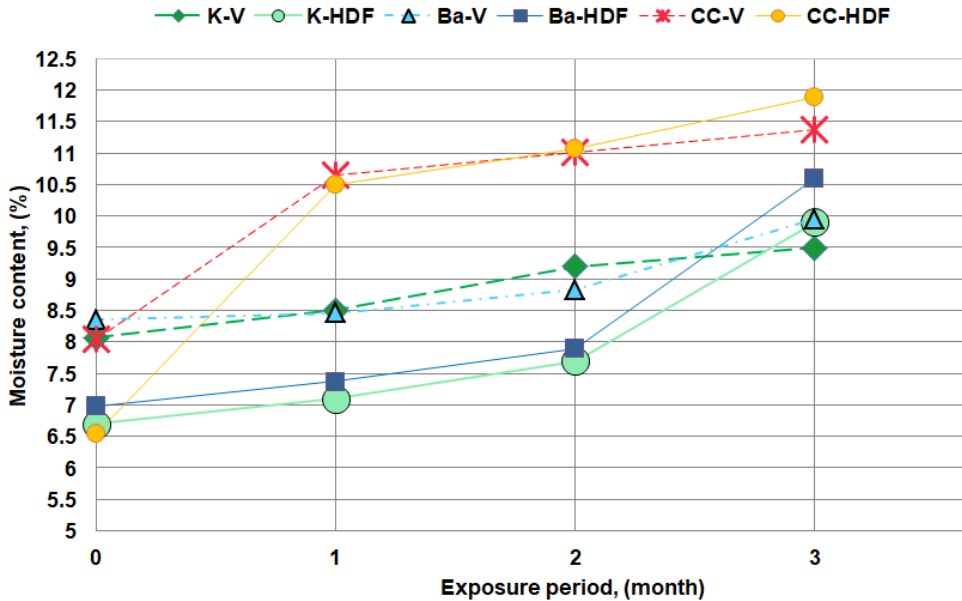

**Figure 4.** Variation of MC for all panels in all three situations of exposure.

The panels exposed in the climatic chamber (Table 5) exhibited the highest increase of MC: 41% for V blockboards and 82% for HDF blockboards after three months of exposure, followed by those exposed in the bathroom and kitchen. There were small differences in moisture content uptake by the boards with the same structure exposed in the kitchen and bathroom. In the climatic chamber, MC increased rapidly after one month of exposure, then the increase was diminished as a result of reaching equilibrium inside the structure. The increase in MC was higher for the structures with veneered faces in all situations compared to those with HDF faces after two months of exposure. It seems that the higher density of HDF face sheets, at least at the beginning of exposure, limited and slowed down the penetration of moisture due to the high compression of the fibres. After two months of testing, it was noticed that the MC values of the blockboards with HDF face sheets exceeded the values of those with veneer face sheets for all cases of exposure. This can be attributed to the affinity of fibres for taking up water and degrading after exceeding swelling. Other researchers [27] also observed that the higher density of the surface layers of particleboards appeared to be unsuitable for exposure to high air humidity.

### 3.2. Dimensional and Shape Stability

Both dimensional and shape stability, or warping the blockboard samples, caused by changes in RH, were assessed through the measurement of deflection at 15 points. Deflection was defined as the difference value between the thickness values measured for each point and the zero value corresponding to the measurement equipment top plate. The thicknesses were always measured at the same points for the entire exposure period until the experiment's completion.

The following 3D surface charts illustrate the warpage of the blockboards exposed in the three environmental conditions before starting the experiment and after three months when the experiment ended. Additionally, the evolution of deflections can also be observed in these charts.

Figures 5–7 show the deflection values and warpage of the veneered (V) blockboards caused by changes in the RH recorded in the three exposure environments, namely kitchen (Figure 5), bathroom (Figure 6) and climatic chamber (Figure 7). The initial shapes of the panels tested in the kitchen space (Figure 5a) and bathroom (Figure 6a) were concave ones. They turned into convex shapes after the panels were exposed to the humid environment of kitchen and bathroom for three months (Figures 5b and 6b). The same trend was also recorded for the panels exposed in the climatic chamber (Figure 7b). After 3 months of exposure in a humid indoor environment, the highest values of deflection were recorded for the edges and corners of the panels.

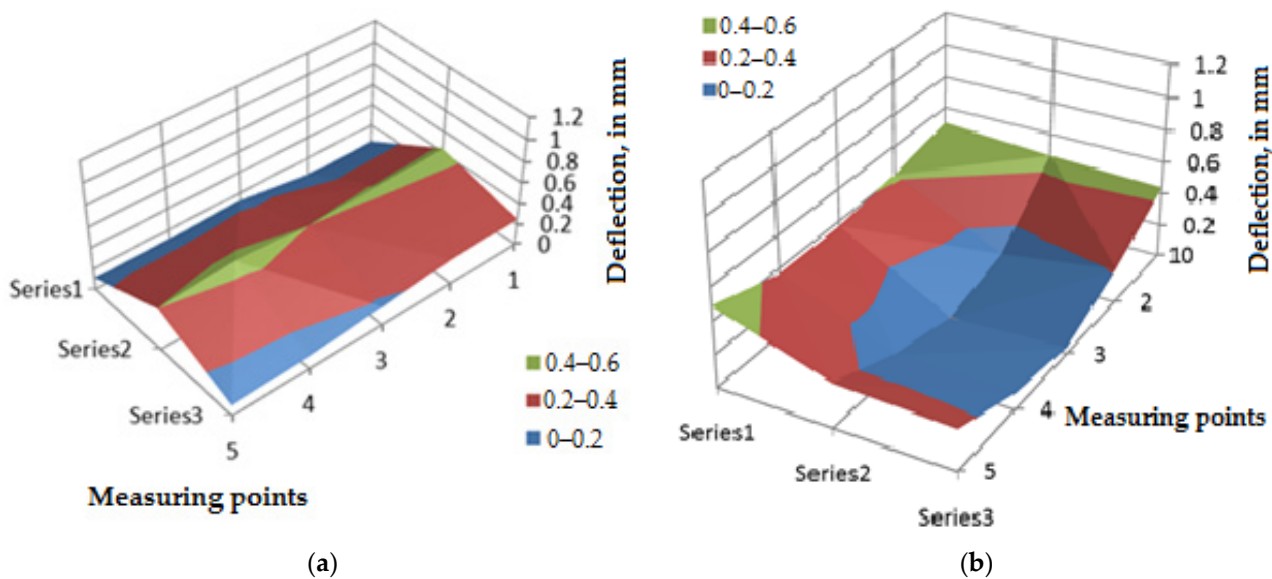

**Figure 5.** Deflection values and warpage of blockboard with veneer (V) faces tested in the kitchen: (**a**) at the beginning of the experiment; (**b**) after three months of exposure in the kitchen.

Due to the frequent cyclical changes in RH between 50% and 90% programmed at 5 days, the panels tested in the climatic chamber absorbed more humidity from the environment than the ones tested in the kitchen and bathroom, where the changes in RH varied only between 25% and 64% for the kitchen and 43% and 88% for the bathroom but with aleatory changes, depending on the use of these spaces. This higher uptake of moisture from the climatic chamber explained the maximum deflection value of 0.59 mm recorded for the blockboards after three months of exposure in this cyclical climate change.

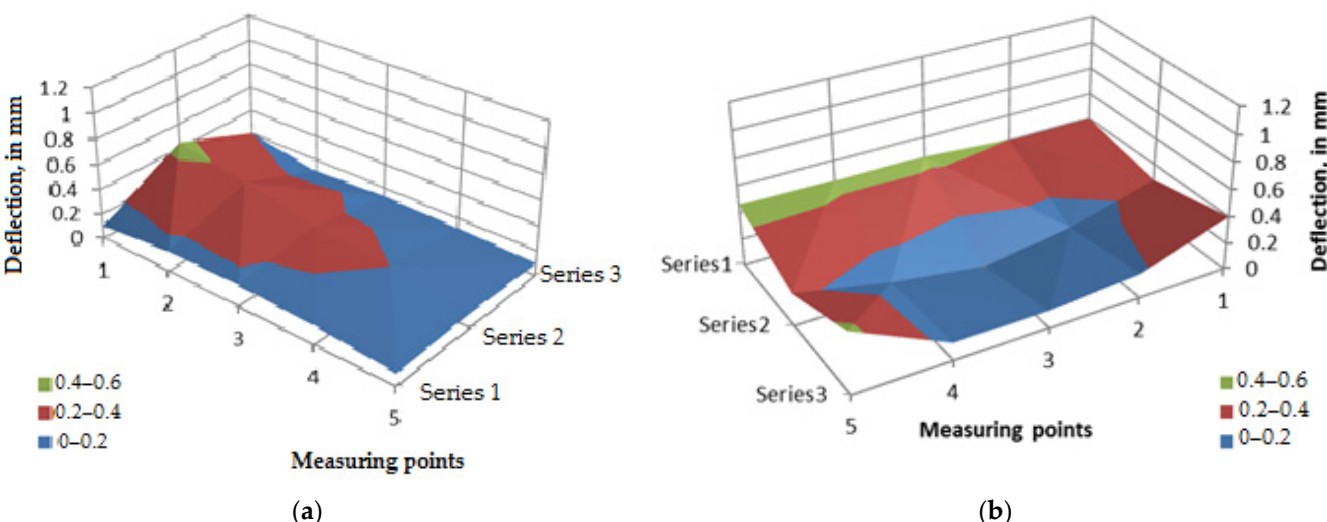

**Figure 6.** Deflection values and warpage of blockboard with veneer (V) faces tested in the bathroom: (**a**) at the beginning of the experiment; (**b**) after three months of exposure in the bathroom.

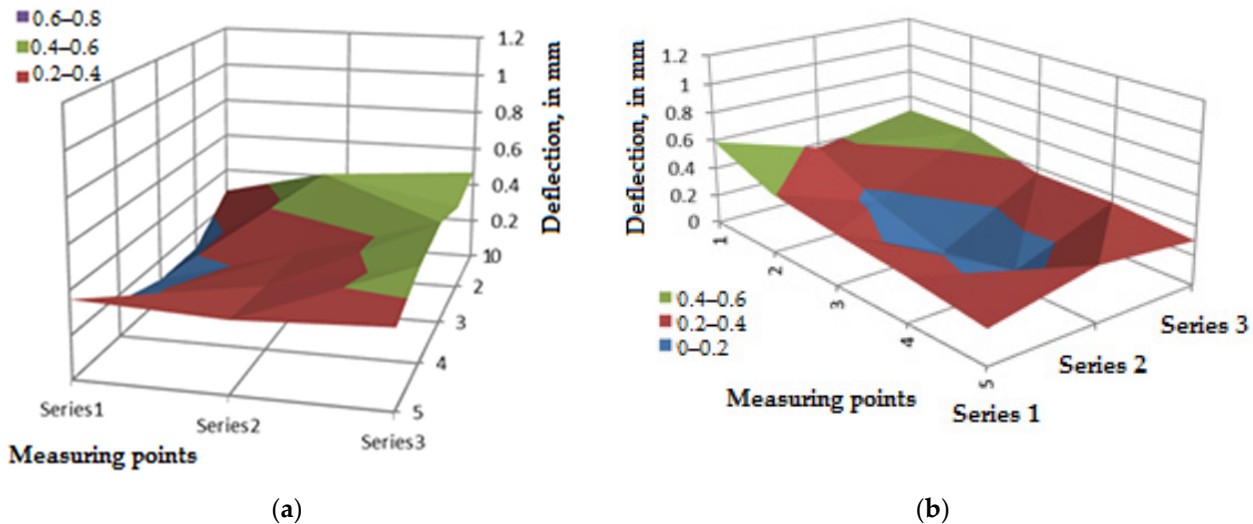

**Figure 7.** Deflection values and warpage of blockboard with veneer (V) faces tested in the climatic chamber: (**a**) at the beginning of the experiment; (**b**) after three months of exposure in the climatic chamber.

The increase of deflection after three months of exposure in the three environmental conditions for these blockboards was no more than 26%. This increase was recorded for the panels tested in the climatic chamber. The warping of these panels was visible as a transformation from concave to convex shape.

Figures 8–10 show the deflection values and the warping of the blockboards with HDF faces caused by changes in the RH recorded in kitchen (Figure 8), bathroom (Figure 9) and climatic chamber (Figure 10). The initial shapes of the blockboards with HDF faces tested in these environments were similar to those with veneer (V) faces, namely concave ones (Figures 8a, 9a and 10a). In contrast to the ones with veneer faces, the blockboards with HDF faces tested in kitchen and bathroom for three months tended to have a flat shape, as seen in Figures 8b and 9b. After 3 months of exposure in a humid indoor environment, the highest values of deflection for these panels were recorded for the corners of the panels. An exception to this rule was observed for the blockboard presented in Figure 10b, exposed to frequent cyclical changes in the RH between 50% and 90% in the climatic chamber, for which the maximum deflection value was recorded in the centre of the panel (1.1 mm). This

recorded thickness variation value is more extreme than in the case of blockboard made from MDF laths for the core and face veneers, where the value of thickness variation was of 0.88 mm [10]. The increase of deflection values after three months of exposure in the three environmental conditions of the blockboards with HDF faces was 162% for the panels tested in the climatic chamber, 152% in the bathroom and 138% in the kitchen. These increases were definitely higher for the blockboards with HDF faces than for those made with veneer faces. These findings are in line with the studies made by other researchers [28], who concluded that the orientation of the veneer grains perpendicular to the core solid wood grains offers a better dimensional stability of the wood composite panels with sandwich structures. This is also the case for plywood, where this arrangement of the veneer layers reduces the warping of the plywood sheets [29]. The fibrous and isotropic structure of HDF face layers causes a uniformly deep and gradual moisture penetration into the structure, and, thus, the warpage of the panels tends to be stabilised after long, continuing, cycling times of changing RH, which is in agreement with [10].

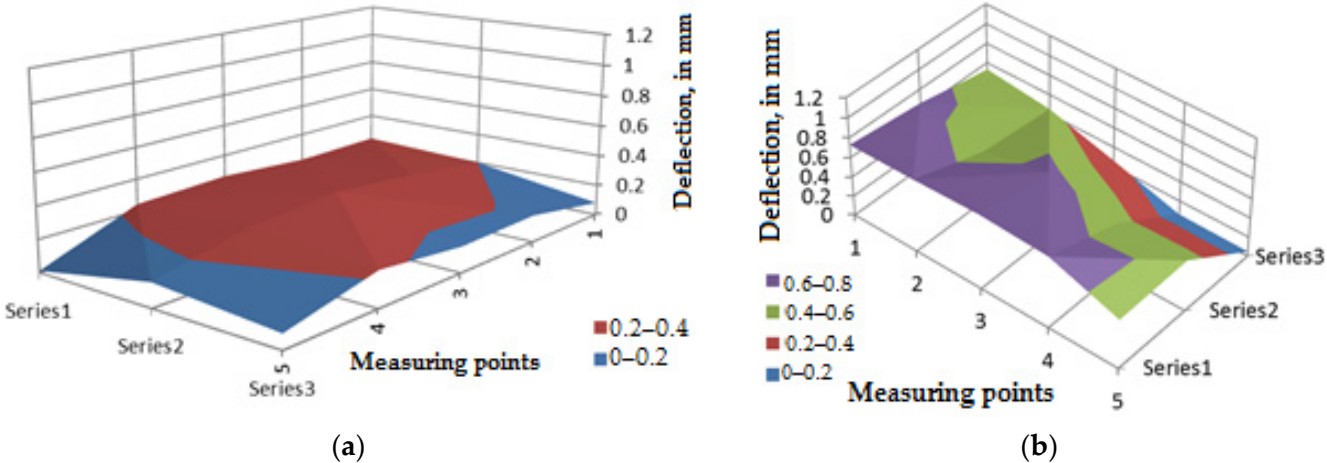

**(a)**  **(b)**

**Figure 8.** Deflection values and warpage of blockboard with HDF faces tested in the kitchen: (**a**) at the beginning of the experiment; (**b**) after three months of exposure in the kitchen.

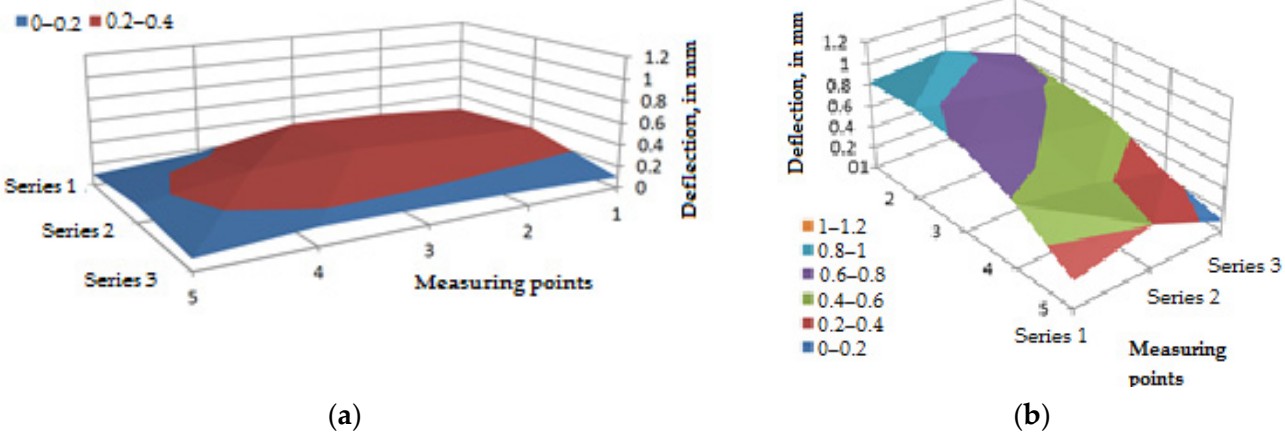

**(a)**  **(b)**

**Figure 9.** Deflection values and warpage of blockboard with HDF faces tested in the bathroom: (**a**) at the beginning of the experiment; (**b**) after three months of exposure in the bathroom.

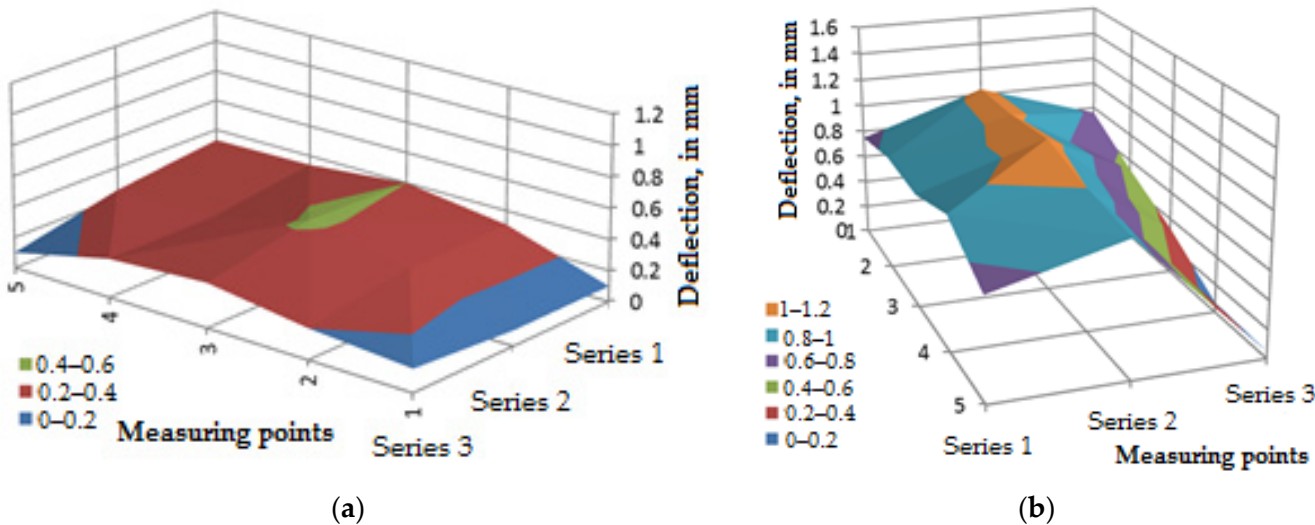

**Figure 10.** Deflection values and warpage of blockboard with HDF faces tested in the climatic chamber: (**a**) at the beginning of the experiment; (**b**) after three months of exposure in the climatic chamber.

### 3.3. Mechanical Properties

The results of the mechanical properties of the tested panels are presented in Table 6. Initial values (before testing the boards) of MOR and MOE were close to those mentioned in the literature for blockboard exposed under stable indoor conditions [30,31]. For the majority of tested boards, higher MOR initial values were registered for blockboards with HDF face sheets (Table 6). After one month of exposure, the flexural properties started to decrease by 1% until 7.5%, with no noticeable differences between the panels exposed in the kitchen and bathroom. A slight effect of moisture content on MOR and MOE after one month of exposure was observed for both blockboard types. In contrast, the exposure in accelerated climate (CC) determined a decrease of strength after one month, within the range of 4% and 13%, more evident for blockboards with HDF faces.

**Table 6.** Mechanical properties of the blockboards exposed in different climate conditions (average values).

| Time of Exposure | Kitchen | | Bathroom | | Climatic Chamber | |
|---|---|---|---|---|---|---|
| | MOE, [N/mm$^2$] | MOR, [N/mm$^2$] | MOE, [N/mm$^2$] | MOR, [N/mm$^2$] | MOE, [N/mm$^2$] | MOR, [N/mm$^2$] |
| Structure 1, with veneer face sheets | | | | | | |
| Before exposure | 6820 | 44.1 | 7120 | 43.6 | 7133 | 48.2 |
| | (620) | (5.5) | (450) | (2.1) | (550) | (5.4) |
| 1 month | 6750 | 42.2 | 6820 | 41.8 | 6830 | 45.6 |
| | (570) | (4.2) | (660) | (2.7) | (733) | (4.5) |
| 2 months | 6440 | 37.4 | 6680 | 38.6 | 6223 | 41.6 |
| | (540) | (3.9) | (450) | (3.2) | (426) | (5.7) |
| 3 months | 6230 | 36.5 | 6470 | 32.1 | 5665 | 32.8 |
| | (230) | (2.2) | (340) | (4.2) | (399) | (1.9) |
| Structure 2, with HDF face sheets | | | | | | |
| Before exposure | 6380 | 48.3 | 6860 | 49.2 | 6630 | 44.3 |
| | (510) | (5.3) | (678) | (3.6) | (613) | (3.7) |
| 1 month | 5950 | 46.8 | 6530 | 45.5 | 5783 | 40.2 |
| | (420) | (4.5) | (546) | (1.4) | (733) | (4.3) |
| 2 months | 5746 | 43.5 | 6080 | 38.4 | 5365 | 36.4 |
| | (254) | (5.6) | (453) | (2.7) | (486) | (4.8) |
| 3 months | 5480 | 38.2 | 5746 | 34.2 | 4700 | 28.7 |
| | (240) | (3.6) | (380) | (2.5) | (385) | (2.1) |

Values in the parenthesis are standard deviations.

The boards exposed in the climatic chamber had an accentuated decrease of the mechanical properties compared to the panels exposed in the kitchen or bathroom. The increase of moisture content in the structures caused the reduction of bending strength and modulus of elasticity by approximately 32% and 21% for V blockboards and by approximately 35% and 20% for HDF blockboards.

The higher decrease of mechanical properties was observed on blockboards with HDF face sheets compared to those with veneers due to the highest MC uptake. The noticeable decreases of the MOR and MOE values over 65% relative air humidity are in line with the results obtained by other researchers [17], who observed that these reductions are more intensive for fibrous panels of higher density when the relative air humidity is raised from 65% to 95%. A similar tendency for strength reduction for different wood-based composites when MC increases was also found by other researchers [32–35].

### 3.4. Surface Soundness

Surface soundness (SS) testing results are presented in Figure 11. EN standards do not present any requirements for blockboards, but, generally, the industry requirement for MDF flooring is 1.2 N/mm$^2$, and, for particleboard (PB) designed for interior use, this requested limit is 0.80 N/mm$^2$ (the green line from Figure 11) [36].

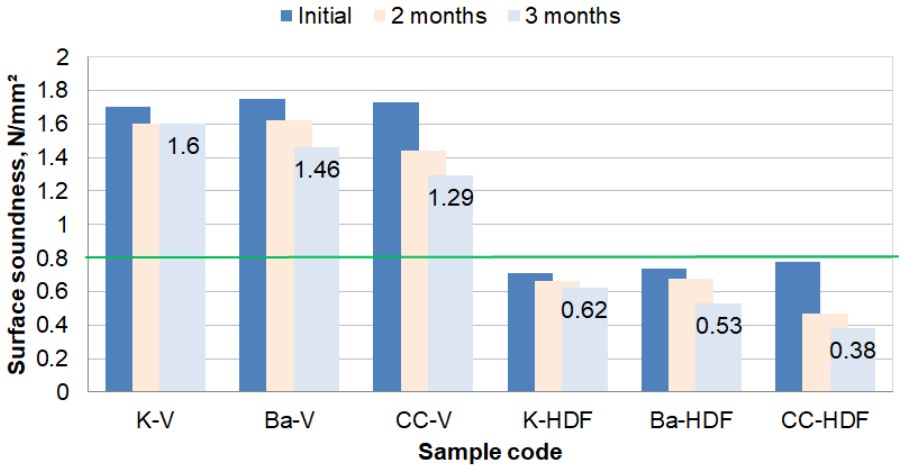

**Figure 11.** Surface soundness results.

Initial (before exposure) SS for all samples was around 1.7 N/mm$^2$ for V blockboards and below 0.8 N/mm$^2$ for HDF blockboards. The higher the density of material, the higher SS expected. In the present research, the blockboards with HDF face sheets (with high density) had the lowest SS values compared to blockboards with lower density of the faces (V blockboards). This unusual behaviour was found also by other researchers for HDF [37]. As can be seen in Figure 11, the SS decreased as the MC increased. At the highest uptake of humidity, which was obtained in the climatic chamber, the tested boards had the lowest SS after three months of exposure: 1.29 N/mm$^2$ for V blockboards and 0.38 N/mm$^2$ for HDF blockboards. The climate conditions in the kitchen had a slight influence on SS, which decreased by approximately 6% for V blockboards and 12% for HDF blockboards. The effect of high values of RH occurred in bathroom space and in climatic chamber, and it was more pronounced after three months of exposure, determining the increase of SS by 25% for Ba-V board, 28% for Ba-HDF board, 25% for CC-V board and 51% for CC-HDF board. All surfaces of the face sheets were visibly affected by the SS test after three months of exposure to RH variations. The surface layer was affected more by the SS test when the moisture absorption increased, without affecting the entire structure.

## 4. Conclusions

The results of this research show that exposure to variation of relative humidity (RH) affects the properties of blockboards. Occasionally, the variation of RH between 25% and 64% (kitchen) had a slight influence on physical (thickness, weight, moisture content and warping) and mechanical properties (MOR, MOE and SS). The effect on properties was more pronounced when RH rose to values between 65 and 90% (bathroom and climatic chamber), causing uptake of moisture, and was different on the two structures.

In the climatic chamber, the dynamics of the parameter variations were different than in the other two environment climates because of the programmed cyclic changes in RH, which reached a maximum value of 90%. RH impact was also observed on the thickness swelling and variations of weight, more pronounced on the panels exposed in climatic chamber, followed by those tested in the bathroom and kitchen. Exposure to different levels of RH affected the blockboards with HDF face sheets more, in terms of warping, compared to those with V face sheets. This might be explained by the orientation of the veneer grains perpendicular to the core solid wood grains, as also concluded by other researchers [28].

The higher decrease of mechanical properties (MOR, MOE, SS) was observed in the HDF structures, which recorded the highest MC at the end of exposure period compared to the initial state. It might be attributed to the transverse direction of the veneer, which inhibits the deeper moisture penetration compared to the fibrous structure of the HDF face layers. The blockboards are suitable to be used for interior applications with the recommendations of avoiding long-term exposure to RH over 65%.

**Author Contributions:** Conceptualization, O.Z.; methodology, O.Z. and C.C.; validation, O.Z and C.C.; formal analysis, O.Z. and C.C.; investigation, O.Z.; data curation, O.Z. and C.C.; writing—original draft preparation, O.Z.; supervision, C.C. and O.Z.; project administration, O.Z.; funding acquisition, O.Z. All authors have read and agreed to the published version of the manuscript.

**Funding:** This research received no external funding.

**Acknowledgments:** We hereby acknowledge the structural funds project PRO-DD (POS-CCE, O.2.2.1., ID 123, SMIS 2637, No. 11/2009) for providing the infrastructure used in this work and the contract no. 7/9.01.2014.

**Conflicts of Interest:** The authors declare no conflict of interest.

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
