# Peer review of "Effect of Variable Conditions of Exposure on the Physical and Mechanical Properties of Blockboards"

_applsci, doi:10.3390/app12020609_

Round 1

Reviewer 1 Report

  • The introduction needs to be improved, the literature review section needs to be enhanced. There are many recent papers on the subject matter that needs to be added
  • Lines 41 – 42, the paragraph is too short and not meaningful
  • The problem statement and research gap needs to be improved
  • The research objective needs to be rewritten
  • The materials properties needs to be added
  • The conclusions can be rewritten in number or bullet formats and should be based on the objectives

Author Response

Article

Manuscript ID: applsci-1548891

Title:  Effect of variable conditions of exposure on the physical and 2 mechanical properties of Blockboards

Authors: Octavia Zeleniuc, and Camelia CoÈ™ereanu   

Reviewer 1

ï‚·  The introduction needs to be improved, the literature review section needs to be enhanced. There are many recent papers on the subject matter that needs to be added

ï‚·  Lines 41 – 42, the paragraph is too short and not meaningful

ï‚·  The problem statement and research gap needs to be improved

ï‚·  The research objective needs to be rewritten

ï‚·  The materials properties needs to be added

ï‚·  The conclusions can be rewritten in number or bullet formats and should be based on the objectives

The authors response

Thank you the reviewer for all the recommendations.

Please, find our modifications, as follows:

  1. The introduction needs to be improved, the literature review section needs to be enhanced

The authors’ response: Lines 43-68 were modified by adding new references and new comments, as follows:

In recent years, there has been a growing demand for wood and the price of logs has experienced a huge increase. For this reason, some of the timber industry has been forced to efficiently use the existing wood resources, including the wood with defects or small sized wood resulted in the wood processing, fast growing wood species (Singh 2020) or recycled wood, such as post-consumer wood (Gayda 2020) or from the construction sites [3 – Texeira].

In the furniture industry, there has been an increase in the demand for the manufacture of blockboards, due to its structural and strength advantages. Blockboard is preferred over plywood for furniture manufacturing, especially in applications such as long shelves, top tables and side panels, because of the high resistance to warping or twisting and durability. Blockboard is a highly appreciated sustainable material, used mainly for furniture, combining its durability with light weight. The actual research works aim to find solutions to lower the weight of blockboard by using light cores from resinous wood with various machined profiles and resistant face layers (Rozins 2020). The blockboard panels are preferred by the furniture manufacturers because they can replace some solid wood elements with this material, lowering the product weight and maintaining the same properties and quality. Blockboard is also an ecological material with very low formaldehyde emission level (Böhm, 2012 , Haixia 2015,).   

The most studies were oriented to the study of physical and mechanical performances of blockboards constructed with various cores and faces. A part of the research works are oriented to the possibility of recycling wood resulted from the construction sites for the core of blockboard with and without bonding, investigating the influence of core material on the mechanical properties of the boards [3, Bellevile 2011]. Innovative method without adhesive, such as dowel welding of blockboard core strips with 0º-20º dowel insertion angle resulted in good mechanical properties of the boards (Bellevile 2011). A part of the studies investigates new designs of the structures [4,5] and various raw materials for the core [6-8].  Other studies were oriented to the joint types used for the core stripes, to their lengths and their influence on the properties of blockboards [9,10, Bellevile 2011].

There are few studies related to blockboard properties and structures. The influence of moisture content or of the relative humidity of air on the physical and mechanical properties of wood based composites were also in the attention of researchers, but only in case of particleboards and fibreboards [11-13]. There are also studies with regard to weathering stability and durability of plywood [14], but no similar studies refer to blockboards and some on lightweight stabilised  blockboard, newly introduced product on the market, that show a decrease in internal bond and bending strength with the increase of RH from 30% to 85% (Rozins 2020).    

  1. Lines 41 – 42, the paragraph is too short and not meaningful

Line 38-42: We modify the sentences as follow: Total production of plywood, blockboards and LVL together represented 40% of all wood-based panel production in 2018 for a production of 163 million m³ [1], from which 145 000 m3 represents only the blockboards production of Holzindustrie Schweighofer, the world’s largest blockboard plant [2].

  1. The problem statement and research gap needs to be improved. The research objective needs to be rewritten.

Lines 70-73: The authors made accordingly the modifications, as presented below :

The present research aims to analyze the effect of variable environmental conditions on the physical and mechanical properties of blockboards, in order to assess the drawbacks of using this material in spaces where an increase in humidity could influence their stability and mechanical strengths. Two types of blockboards (with veneer and HDF faces) were exposed in three different climate conditions (in the kitchen, bathroom spaces and in a programmed climatic chamber environment), and after three months, their behaviour was evaluated. The surface warpage through the evolution of deflection was also evaluated.

  1. The materials properties needs to be added

Line 89: The following data were added:

Blockboard boards from which the specimens were cut for the experiments had the following characteristics: board format: 2500x1250 mm; moisture content: 8-12%; thickness tolerance: + 0,2 mm - 0,6 mm; emission class: E1.

  1. The conclusions can be rewritten in number or bullet formats and should be based on the objectives

The authors thank the reviewer and emphasize that the conclusions follow the instruction for authors were are not specify to use bullets or numbers, and all the issues formulated in the objective section are discussed in the conclusion.

Reviewer 2 Report

Dear Authors,

It is an interesting article and some of the readers of this journal would be interested in material presented in the work. Below you can find a list with some minor revisions. Once it is revised, I guess manuscript will be ready for publication.

  • Line 159 and line 161

Please clarify the abbreviation “days*2” in the text.

  • Line 192, line 195, line 199 and line 201

What do you mean “TS values”?

  • Figure 5 (a)

Please check the quality on the left image (maybe load a higher quality version of this image).

  • Line 302

Chapter headline “3.2. Mechanical properties” change to “3.3. Mechanical properties”

Best Regards.

Author Response

Article

Manuscript ID: applsci-1548891

Title:  Effect of variable conditions of exposure on the physical and 2 mechanical properties of Blockboards

Authors: Octavia Zeleniuc, and Camelia CoÈ™ereanu  

Reviewer 2

It is an interesting article and some of the readers of this journal would be interested in material presented in the work. Below you can find a list with some minor revisions. Once it is revised, I guess manuscript will be ready for publication.

The author’s response

The authors thank the reviewer for all the recommendations and appreciation.

Please, find our modifications as follow:

  1. 1. Line 159 and line 161

Please clarify the abbreviation “days*2” in the text.

The authors response:

Lines 138-139:  We modified the paragraph as follow: These parameters were measured twice a day (days*2) for all the period of 90 days of testing and their variations for kitchen and bathroom conditions of exposure are presented in Figure 3.

  1. 2. Line 192, line 195, line 199 and line 201

What do you mean “TS values”?

Line 192: We added:  thickness swelling (TS)

  1. 3. Figure 5 (a)

Please check the quality on the left image (maybe load a higher quality version of this image).

We have checked and replaced the image as suggested.

  1. Line 302

Chapter headline “3.2. Mechanical properties” change to “3.3. Mechanical properties”

Thank you the reviewer. We modified as suggested.

Reviewer 3 Report

A brief summary – The authors followed and analized physical and mechanical properties of Blockboards at variable conditions of exposure to temperature and relative humidity of air.

They carried out an extensive study of parameters according to the European standards. The results are of high practical value for the producer of Blockboards and important for interested society.

General Blockboard is widely used in the production of furniture, so there is a need to know how various environmental conditions influence its properties.

The present study represents a detailed and through study of these influences. The authors performed thorough investigation of samples using relevant characterisation techniques. The figures and tables properly show the data and are consistent throughout the manuscript. Similar studies are only reported for particleboards and fibreboards, however no studies are available for blockboards.

The manuscript is well-structured and relevant for the field. The conclusions are consistent with experimental evidence. The cited references are current, a number of them within the last 5 years. The authors did not included self-citations.

Author Response

Article

Manuscript ID: applsci-1548891

Title:  Effect of variable conditions of exposure on the physical and 2 mechanical properties of Blockboards

Authors: Octavia Zeleniuc, and Camelia CoÈ™ereanu  

Reviewer 3

A brief summary – The authors followed and analized physical and mechanical properties of Blockboards at variable conditions of exposure to temperature and relative humidity of air.

They carried out an extensive study of parameters according to the European standards. The results are of high practical value for the producer of Blockboards and important for interested society.

General  Blockboard is widely used in the production of furniture, so there is a need to know how various environmental conditions influence its properties.

The present study represents a detailed and through study of these influences. The authors performed thorough investigation of samples using relevant characterisation techniques. The figures and tables properly show the data and are consistent throughout the manuscript. Similar studies are only reported for particleboards and fibreboards, however no studies are available for blockboards.

The manuscript is well-structured and relevant for the field. The conclusions are consistent with experimental evidence. The cited references are current, a number of them within the last 5 years. The authors did not included self-citations.

The authors would like to thank the reviewer for all the comments and appreciation.
